# Low Energy Renovation of Social Housing: Recommendations on Monitoring and Renewable Energies Use



**Bianca Seabra** [1] , **Pedro F. Pereira** [2,*] , **Helena Corvacho** [2] , **Carla Pires** [3] **and Nuno M. M. Ramos** [2]

1   Faculty of Engineering (FEUP), University of Porto, 4200-465 Porto, Portugal; up201504050@fe.up.pt
2   CONSTRUCT (LFC), Faculty of Engineering (FEUP), University of Porto, 4200-465 Porto, Portugal; corvacho@fe.up.pt (H.C.); nmmr@fe.up.pt (N.M.M.R.)
3   Gaiurb EM, 4400-012 Vila Nova de Gaia, Portugal; cpires@gaiurb.pt
*   Correspondence: fpfp@fe.up.pt

**Abstract:** Social housing represents a part of the whole building stock with a high risk of energy poverty, and it should be treated as a priority in renovation strategies, due to its potential for improvement and the need to fight that risk. Renovation actions are currently designed based on patterns that have been shown to be disparate from the reality of social housing. Thereby, a monitoring study is essential for the evaluation of the actual conditions. An in-depth characterization of a social housing neighborhood, located in the North of Portugal, was carried out. Indoor hygrothermal conditions were analyzed through a monitoring campaign. It was possible to identify the differences in indoor conditions of the dwellings and understand the influence of occupancy density and occupants' behavior. In order to identify the actual occupancy and the type of use, a social survey was performed. A renovation action will soon take place, and a monitoring and survey plan is proposed for the post-renovation period, based on a previous evaluation of the renovation impact, using DesignBuilder software and the real occupancy profiles. In social housing context, since energy consumption for heating and cooling is punctual or non-existent, the focus of low energy renovation should be based on passive strategies that reduce the energy demand. The remaining energy needs should be supplied by renewable energy sources, reducing energy poverty, and enhancing quality of life.

**Keywords:** social housing; renovation; monitoring plan; building energy simulation; energy-efficiency; thermal comfort

## 1. Introduction

Urban areas are constantly growing, both in size and in population, leading to the emergence of new challenges related to demography, climate change, energy consumption, and transport, among others [1]. One of the most relevant factors refers to the growing energy needs and, consequently, the association of new problems such as energy poverty, environmental damage, and the increasing vulnerability of people against these phenomena [2], compromising the resilience of urban systems and social well-being [1].

Energy consumption in buildings in the European Union (EU) is responsible for 40% of energy consumption and 36% of greenhouse gas emissions [3], and identical percentages were quantified in North America [4]. In southern European countries, the percentage of the buildings energy consumption decreases to 30% of the total [5]. Furthermore, the directive 2018/844/EU [6] estimates that around 35% of EU buildings are over 50 years old and almost 75% of the built stock in the European Union is energy inefficient. The same directive also highlighted that the renovation of the existing buildings has the potential of reducing the energy consumption by 6% and reducing the $CO_2$ emissions by 5%. Moreover, improved energy efficiency in buildings is a way to reduce the energy bills and, therefore, reduce energy poverty, improving building occupants' indoor environmental quality (IEQ) and general well-being [7,8]. Thus, the continuous improvement of the energy efficiency

of buildings plays a fundamental role in reaching the goal of carbon neutrality by 2050, established in the European Green Deal [6]. However, only about 1% of the buildings is renovated each year, with member states' rates varying from 0.4% to 1.2%. To meet the established climate and energy objectives, current rates of renovation must be at least the double [6]. Therefore, it becomes essential to invest in the renovation and maintenance of buildings in order to improve environmental, economic, and social aspects, enhancing the built heritage, which is reflected in the improvement of the population's quality of life [9]. EU awareness of the urgency of this matter is because in 2017, 10.2% of the households in the EU spent more than 40% of their income on housing costs, but this percentage increases to 37.8% when considering households in poverty risk, which represents a number above 150 million people [10]. Despite a slight decrease in the housing overload rate in the last two years [10], the population still feels scarcity of affordable housing, as their price is rising faster than household income [11]. Hence, there is a high demand for social housing or controlled prices, framed in the financial capacities of the underprivileged inhabitants [11,12].

Currently, the share of social housing in Portugal corresponds to 2% of the total housing stock [10,13], corresponding to 119,691 dwellings [14], which provides housing to approximately 113,000 households, representing approximately 270,000 individuals (2.5% of the Portuguese population) [13]. According to National Institute of Statistics report [14], it is concluded that a large number of people still live in precarious housing, with 187 municipalities being flagged as having a housing shortage. Furthermore, 25,762 families were identified as being in a situation of unsatisfactory habitability (0.78% of the total families residing in these municipalities), and 14,748 buildings and 31,526 dwellings are without minimum living conditions. It is also concluded that the greatest concentration of this phenomenon is found in the metropolitan areas of Porto and Lisbon, where a total of 74% of the families in that situation is registered.

In 2018, according to a survey conducted in the European Union, 7.3% of the population did not have the capacity to be able to heat their housing in a convenient way [15]. However, we have to consider that this value is an average, and if we analyze the situation in each member state, these numbers may be much higher. The largest number of people who found that they were unable to keep their home adequately heated was registered in Bulgaria (34%), followed by Lithuania (28%), Greece (23%), Cyprus (22%), Portugal (19%), and Italy (14%) [15].

Based on the observed works, which focus on the analysis of social housing in Portugal [8,16–20], it can be concluded that this housing portion has a very low energy consumption for heating and cooling, due to several problems: the fact that a majority do not have centralized heating or cooling systems, energy poverty, a housing stock with low energy performance, and high energy prices. Putting together the aforementioned with the milder winter climate conditions of the country, the current legislation is not adapted to this type of sector, with usage patterns and consumption much higher than the real ones [21]. Consequently, a financial effort is needed to rehabilitate the social housing park, in order not only to improve the performance of buildings, but also to strengthen social cohesion [22]. The specificities of the social housing distinguish them from all other types of housing, and the definition of universal thermal comfort standards is questionable, as different vulnerability patterns must be recognised, taking into account social and cultural differences [16]. Therefore, customized and comprehensive studies should be made in social housing renovations works in the areas of IEQ and energy consumption [8,21,23,24]. Other approaches are also being applied in the context of social housing. Lucchi and Delera [25] used an occupant-centered design-driven approach to deep refurbishment and revitalization of a neighborhood in Italy. The project designers selected the retrofit solutions based on the local social–economic conditions given by participatory actions. In Portugal, the participatory action in architecture began in 1974 and has been applied since in many social houses [26].

It is essential to know the real conditions in which the buildings that will undergo renovation actions are used, so that the actions result as expected. Beyond the knowledge of the buildings' passive and active systems, the specificities of the occupants and households of social housing must be known prior to any renovation work. Furthermore, the setup of monitoring plans and subsequent realistic dynamic simulation of buildings may reduce performance gaps between the simulated and the measured after the renovation takes place. Currently, the post-occupancy monitoring of social housing in Portugal is not widely applied. This work tries to fill this gap with the monitoring of a building prior to its renovation. The main objectives of this work are:

- Propose a monitoring plan to study the pre-renovation conditions in terms of hygrothermal conditions;
- Propose a monitoring and a survey plan to study the effectiveness of the renovation works in terms of thermal comfort and energy consumption;
- Propose renewable energies customized for the social housing specificities.

## 2. Methodology

The present work was carried out within the scope of the INTERREG SUDOE project ENERGY PUSH (SOE3/P3/E0865) and contributes to the project aim of developing a building management system for social housing. It should be capable of measuring the effect of the retrofit works, analyzing the necessity for new interventions in the future, and analyzing the IEQ and the necessity of adjustments in ventilation strategies and renewable energies.

The present study comprises three phases. The first includes the elaboration of a monitoring plan in order to obtain a characterization of the hygrothermal conditions inside the houses. Thus, through a field and social research, it is intended to demonstrate the initial conditions verified in terms of the energy performance of buildings and the thermal comfort of the occupants. To achieve that, the architectural features of the selected neighborhood were studied in its current state, and the planned renovation measures were analyzed. Then, a monitoring campaign and a social survey were carried out: the used measurement equipment and the strategies for placing it on site were carefully defined, and the social survey was setup focused on the specificities of social housing. The monitoring and the survey results were evaluated in order to get a realistic description of the field conditions and to identify the most relevant influent factors. Subsequently, the second phase uses the building energy simulation program Design Builder to model the occupants' profiles, identified in the pilot monitoring/surveys and inserting the projected renovation solutions. The aim is to expose the expected and adjusted results of the intervention and the benefits that come from it, and based on the results analysis propose a post-renovation monitoring plan. Finally, bearing in mind that the renovation projects are produced based on legislation that starts from theoretical assumptions that are not always verified, mainly in social housing, it is also intended to achieve a deeper understanding of the conditions experienced in this type of housing. This will bring awareness to the need to develop more detailed and adapted projects, using tools such as sensors and dynamic simulation software to improve renovation projects and, consequently, the conditions of social housing in Portugal.

### 2.1. Studied Neighborhood

The case study is a Portuguese small social housing neighborhood in the Municipality of Vila Nova de Gaia (Figure 1). The neighborhood, built in 1997, consists of two parallel buildings with three entrances each. Low-income households occupy all the 35 dwellings. Each building has three floors above the ground. Each entrance gives access to two dwellings per floor (six dwellings per entrance). In each floor, there is a two-bedroom apartment and a three-bedroom apartment.

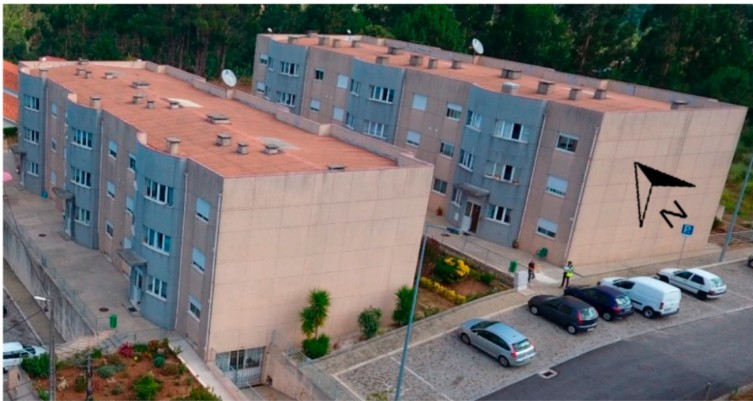 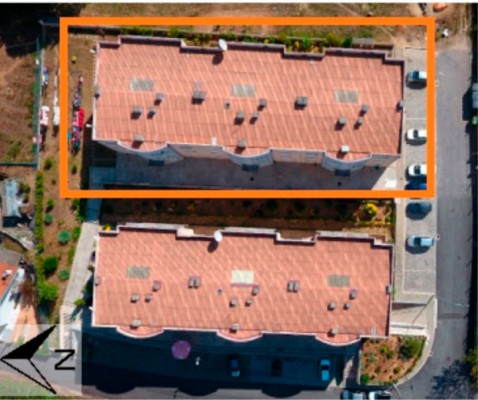

**Figure 1.** Aerial views of the studied social housing neighborhood.

The main façade (laundry and living room) faces west, and both buildings have similar construction solutions (Table 1), with the structure consisting of concrete columns and beams combined with concrete slabs of prestressed joists and ceramic blocks.

**Table 1.** Current construction solutions.

| Construction Element | Thermal Transmittance Coefficient, U-Value (W/(m$^2 \cdot$°C)) | Construction Solution |
|---|---|---|
| External walls (0.40 m) | 0.96 | Double hollow brick masonry wall, 11 and 15 cm thick, plastered on the inside and coated with light coloured industrial cementitious mortar on the outside. |
| Walls that separate apartments from the staircase (0.24 m) | 1.16 | Simple hollow brick masonry wall coated on the outer face with light colored industrial cementitious mortar and on the inner face with stucco or ceramic tiles. |
| Pitched roof | $U_{asc}$ = 1.90 $U_{desc}$ = 1.44 | The second floor ceiling slab is a concrete slab of prestressed joists and ceramic blocks with no thermal insulation. The attic is poorly ventilated. The roof slopes are covered with fibre cement sheets. |
| Ground floor slab (with sanitary air space underneath) | 1.18 | Concrete slab of prestressed joists and ceramic blocks coated with wood parquet or ceramic tiles. |
| Windows | 3.10 | Aluminium frames with clear double-glazing and an outer plastic roller blind of a light colour. |

The dwellings rely on natural ventilation, and they include vertical ducts in the bathrooms. There is no centralized heating or cooling system installed, and gas heaters produce domestic hot water.

The first phase of the planned renovation will only cover the building to the east, marked on Figure 1. The intervention covers 18 of the neighborhood dwellings.

By the start of the present study, renovation measures were already defined and approved by the Municipality, aiming to solve the construction anomalies that arose in the buildings, namely condensation and infiltration of rainwater, promoting an architectural requalification, an improvement of the surrounding urban environment, and a reduction in energy consumption and related costs. The planned renovation measures (Table 2) are based in standard procedures, following the regulation requirements.

**Table 2.** Planned new construction solutions.

| Renovation Action | Construction Solution | U-Value (W/(m$^2\cdot$°C)) |
|---|---|---|
| Application of thermal insulation on the external walls after cracking repair | ETICS | 0.390 |
| Replacement of fibre cement sheets by sandwich panels and application of thermal insulation on the ceiling slab | Sandwich panels as roof covering. Application of 0.10 m of mineral wool on the second floor ceiling slab. | $U_{asc}$ = 0.330 $U_{desc}$ = 0.313 |

### 2.2. Monitoring Campaign

Seven dwellings of the case study were continuously monitored for approximately two months, starting in April and ending in May. Dwellings were monitored in both buildings. Dwellings are numbered from one to seven, and some of their relevant features are given in Table 3.

**Table 3.** Monitored dwellings.

| Dwelling Number | Floor | Glazing Orientation | Occupancy |
|---|---|---|---|
| 1 | Ground-floor | E and W | Occupied |
| 2 | First floor | E and W | Occupied |
| 3 | First floor | E and W | Occupied |
| 4 | First floor | E and W | Occupied |
| 5 | First floor | E and W | Occupied |
| 6 | Second floor | E and W | Occupied |
| 7 | Second floor | E and W | Vacant |

The type of sensor used for the monitoring campaign was the "Sensirion-SHT31 Smart Gadget Development Kit" (Figure 2), which is a sensor of relative humidity and temperature inserted in a simple reference circuit board, with approximately 3.3 × 7.5 × 0.5 m$^3$ and unit weight of 20 g. It can also communicate with an iOS or Android smartphone compatible with the available "My Ambience" application, via Bluetooth Low Energy (BLE). The device stores the measured values in the built-in memory and is powered by battery. The stored data can be downloaded to the application and exported. The temperature accuracy is ±0.3 °C for a range of −40 to 90 °C, and for the relative humidity the accuracy is ±2% for 0% to 100%.

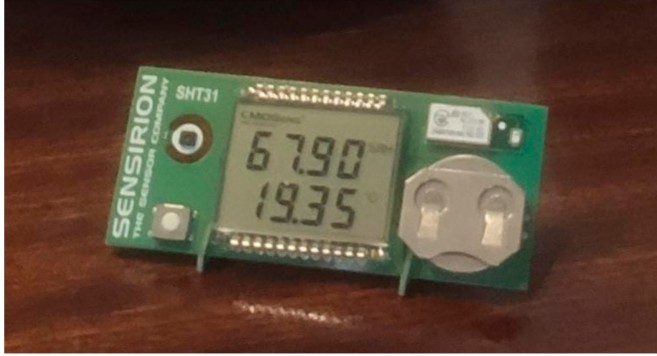

**Figure 2.** Equipment used for monitoring.

The placement of the sensors followed some requirements, based on the documents ISO-7726 [27], Hnat et al. [28], and Pereira and Ramos [29]. The sensor was positioned in the breathable zone; according to EN-16798-1 [30], it represents a part of the occupied

zone at the level of the occupants' head. Regarding the choice of the walls for placing the sensors, priority was given to partition walls, perpendicular to the façades, instead of the external walls, as these ones are influenced by outdoor temperature, and the partition walls that are parallel to a window are more likely to be reached by direct solar radiation. Finally, all sensors were glued to the wall surfaces with double-sided tape so as not to be moved by the occupants or interfered with their daily activities.

### 2.3. Social Survey

In order to characterize the occupancy and the type of use of the dwellings, a social survey was carried out.

The survey included 22 questions with the following main goals:

- Question 1: aims to characterize the neighborhood demographically, in order to get a detailed understanding of households.
- Questions 2 to 10: this section aims to assess the condition of the dwellings concerning the occurrence of construction anomalies, as well as the satisfaction of residents with their home and their perception of comfort in the heating and cooling seasons.
- Question 11: its objective is to define a typical daily time schedule for the occupation of the dwellings.
- Questions 12 to 19: these questions arise as part of the hygrothermal study of the dwellings, since they are meant to characterize the occupants' habits regarding ventilation, heating, and cooling of the dwellings.
- Questions 20 to 22: finally, and based on energy poverty concerns in a social housing context, these questions try to assess the lack of means to afford energy bills and the consequences of it.

### 2.4. Dynamic Simulations

Design Builder software version 6.1.5.004 was used in the dynamic simulation of the building. Design Builder is an interface for Energy Plus and Radiance, which are well-known powerful and globally relied simulation engines.

After modelling the buildings, made easy by the software packages (Figure 3), and defining their occupation and type of use, based on the monitoring and survey data, the simulations were performed and validated, comparing their results to the values measured by the sensors.

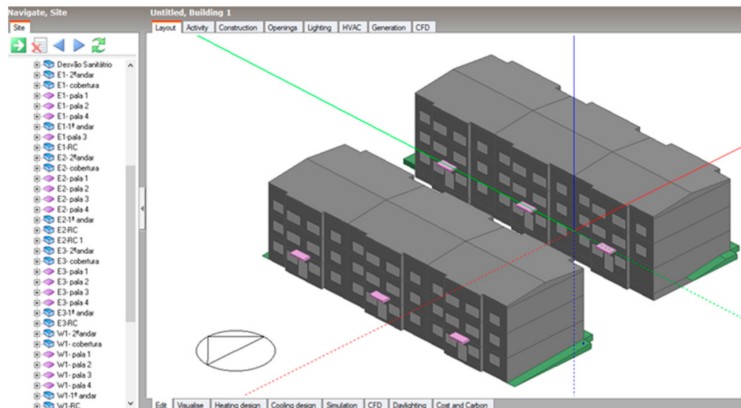

**Figure 3.** Modelling of the buildings of the studied neighborhood, from Design Builder.

After validating the model, some detailed analyses were performed in order to evaluate the relevance of factors such as orientation and location in the building, throughout the year. The final goal of the simulation was to assess the impact of the planned renovation actions and to base the recommendations for a post-renovation monitoring.

## 3. Results

### *3.1. Monitoring Campaign*

#### 3.1.1. Thermal Comfort Analysis

Thermal comfort is a mental condition that shows satisfaction with the thermal environment. Dissatisfaction can be caused by the hot or cold discomfort of the body as a whole or by a change in the perception of unwanted temperature in a specific part of the body. Due to individual differences, it is not possible to specify a thermal environment that satisfies all occupants, and there is always an unsatisfied percentage of people. However, it is possible to specify acceptable environments for a certain percentage of the occupants [31]. Two different approaches for assessing thermal comfort are used in this paper.

According to Portuguese national regulation in the field of energy efficiency in residential buildings, Decree-Law nº 118 (2013) and subsequent Ordinances and Orders [32], a lower limit of 18 °C and an upper limit of 25 °C are considered as a reference, defining a comfort range of indoor temperatures. In the European Standard EN-16798-1 [30], for the cooling season, an adaptive model is suggested, and for the heating season, an analytical method is suggested, with constant values. In Figures 4 and 5, both approaches are shown.

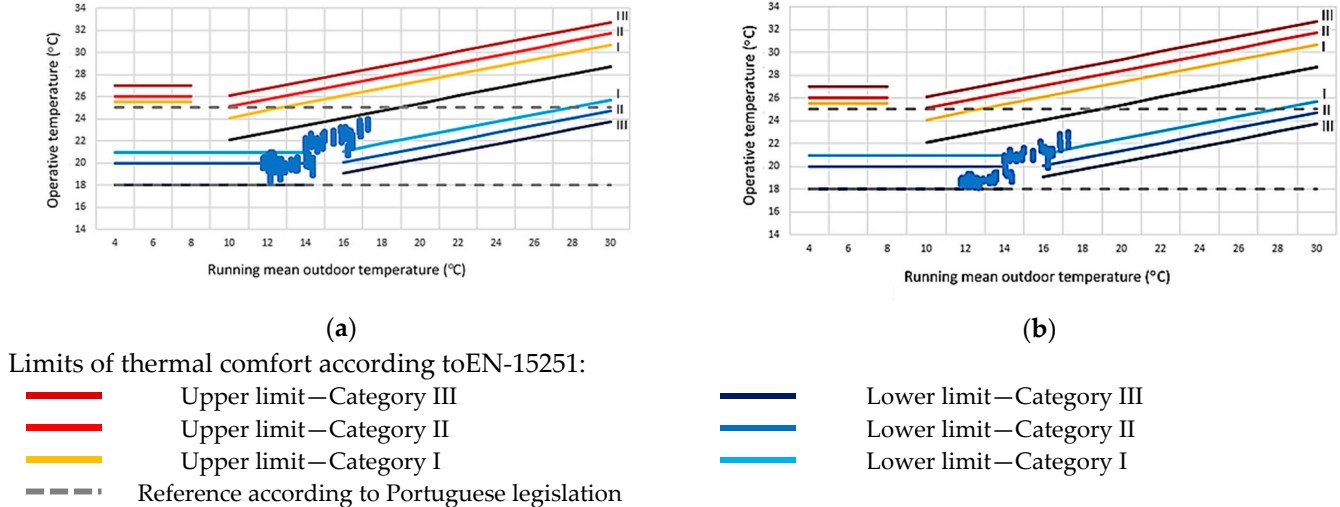

(**a**)                                                  (**b**)

Limits of thermal comfort according toEN-15251:

| | | | |
|---|---|---|---|
| ▬▬ | Upper limit—Category III | ▬▬ | Lower limit—Category III |
| ▬▬ | Upper limit—Category II | ▬▬ | Lower limit—Category II |
| ▬▬ | Upper limit—Category I | ▬▬ | Lower limit—Category I |
| ---- | Reference according to Portuguese legislation | | |

**Figure 4.** Thermal comfort analysis in dwelling Nº5 according to EN-15251 [33]: (**a**) in the living room; (**b**) in the main bedroom.

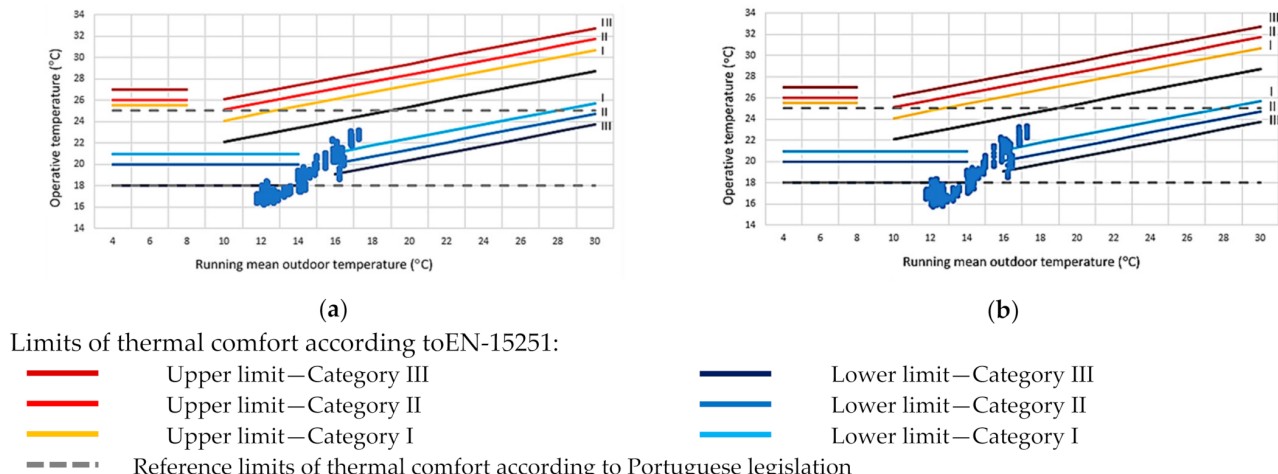

(**a**)                                                  (**b**)

Limits of thermal comfort according toEN-15251:

| | | | |
|---|---|---|---|
| ▬▬ | Upper limit—Category III | ▬▬ | Lower limit—Category III |
| ▬▬ | Upper limit—Category II | ▬▬ | Lower limit—Category II |
| ▬▬ | Upper limit—Category I | ▬▬ | Lower limit—Category I |
| ---- | Reference limits of thermal comfort according to Portuguese legislation | | |

**Figure 5.** Thermal comfort analysis in dwelling Nº6 according to EN-15251 [33]: (**a**) in the living room; (**b**) in the main bedroom.

The comfort analysis focuses on two dwellings, Nº5 and Nº6. These are interesting dwellings, since one is located beneath the roof and the other is overcrowded. The analyzed indoor spaces are the living room (LR) and the main bedroom (R1). The clouds of points in the graphs of Figures 4 and 5 display the pairs of running mean outdoor temperature and indoor temperature measured by the sensors, for the days from 8 April to 21 May. It should be noted that the data for only two months might not be sufficiently representative of the real comfort conditions of the buildings. However, since in the beginning of the measurement period, there were still a considerable number of cold days, and at its end, there were already several quite warm days, it was possible to identify the main trends.

Looking at the graphs, it can be seen that the dwelling Nº6 recorded more hours below the lower limit of category III than the dwelling Nº5. Regarding the maximum allowable temperatures, the values are within the comfort ranges.

In the case of dwelling Nº5, in the living room, the temperature got closer to the upper limits on some days. Especially in the bedroom, there are temperatures below the lower limit of category III. For the remaining indoor spaces, temperatures were within comfort limits.

Despite this analysis, it is necessary to make a critical reflection on these results. For example, in dwelling Nº6, for a running mean outdoor temperature of 16 °C, an indoor temperature of at least 19 °C was registered, and despite being at the lower limit of category III, it is a value that in practice is quite comfortable for the majority of the population. Thus, when using this type of criteria, it is necessary for the occupant to consider a social and cultural context. In the case of the typical Mediterranean climate, it is common for occupants to be more receptive to lower indoor temperatures.

Considering the reference values established by the Portuguese regulation [32], in the LR of the Nº6 dwelling, a total of 515 h below the lower limit of 18 °C was registered, with a maximum difference of 1.7 °C, and in R1, there were 514 h below 18 °C, with a maximum difference of 2.2 °C. There were no values of temperature above 25 °C. Regarding the Nº5 dwelling, the temperatures fell within the comfort range.

### 3.1.2. Analysis of Occupants Influence

According to Ulukavak Harputlugil et al. [34], it is recommendable to use an unoccupied dwelling as a control. This dwelling may serve as a reference, obviously with no influence from the occupants, in comparison to an occupied one. For this comparison, the dwelling Nº7 (vacant) and Nº6 (occupied) were used, both of which are in the same position in the building, but in parallel buildings.

The graphs of Figures 6 and 7 show the comparisons between the measured values of temperature and relative humidity obtained through the sensors, in the dwelling Nº6 (blue curve) and Nº7 (red curve), having as reference the outdoor temperature values (grey curve). The graphs for the living rooms (LR) and main bedroom (R1) are presented, with the values for a reference week (May 1 to 8) for better visualization of the results.

As could be expected, the occupied dwelling has a higher variation over time, in temperature and relative humidity, than the vacant dwelling. The presence of some visible peaks suggests moments where there was interference in the indoor climate and, since the biggest difference between the two dwellings is the presence of occupants, these variations can be attributed to their intervention. Another important observation is the considerable temperature difference in the R1 rooms. This difference, especially from the hottest day, May 3, may be attributed to the occupants' action, such as the use of solar protections that prevented the room of dwelling Nº6 from heating up as much as Nº7's.

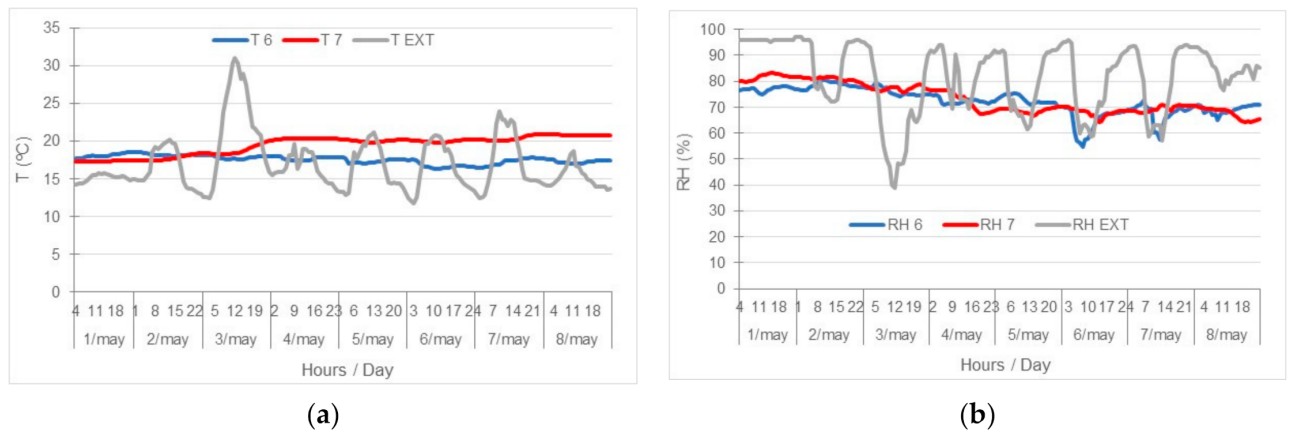

**Figure 6.** A May week comparison between an occupied dwelling (N°6) (blue curve), and a non-occupied dwelling (N°7) (red curve), concerning the conditions in the living room (LR): (**a**) temperature; (**b**) relative humidity.

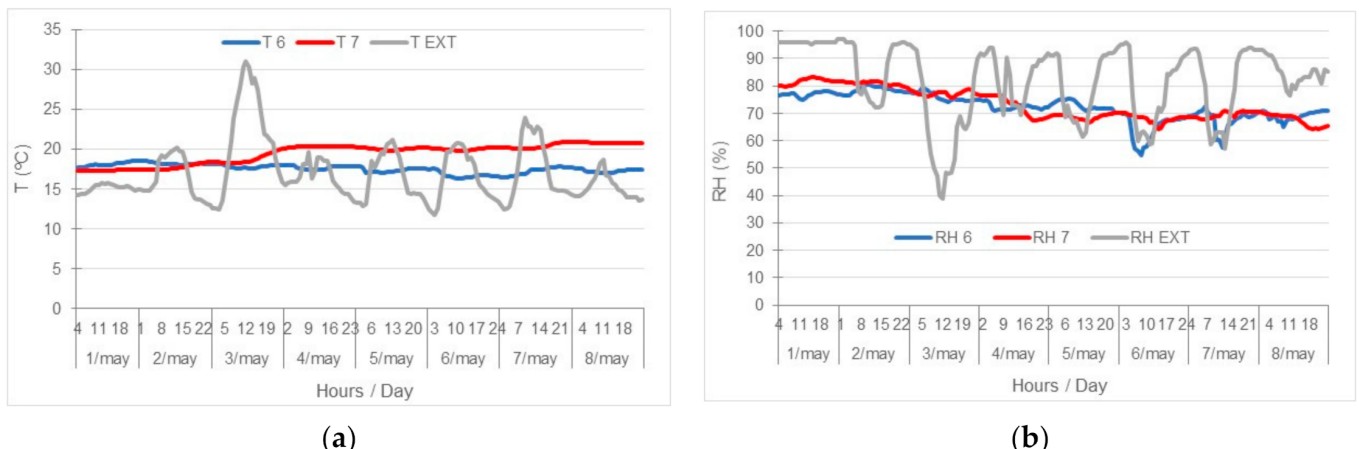

**Figure 7.** A May week comparison between an occupied dwelling (N°6) (blue curve) and a non-occupied dwelling (N°7) (red curve), concerning the conditions in the main bedroom (R1): (**a**) temperature; (**b**) relative humidity.

Two other dwellings of interest are the N°2 and N°5, which, like the previous ones, are in the same position in the building but in parallel buildings. The particularity of these two dwellings refers to their occupancy; both are two-bedroom apartments, but the dwelling N°2 is under-occupied, having only one occupant, while the N°5 has five occupants. The graphs of Figures 8 and 9 show the comparison of the actual temperature and relative humidity, for a reference week (May 1 to 8), between the two dwellings. The overcrowded dwelling has always higher indoor temperatures than the N°2 dwelling, with a maximum difference of 2.9 °C and a minimum difference of 0.7 °C, for the LR, and a maximum difference value of 2.0 °C and minimum of 1.0 °C, for R1. This difference can be attributed to the different ventilation rates and times and occupation density, showing the influence of the different occupation styles.

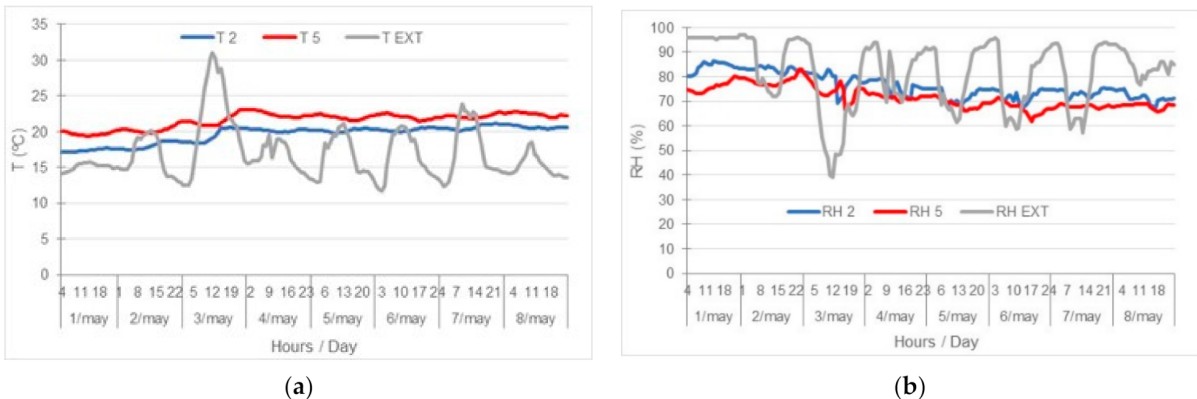

(**a**)          (**b**)

**Figure 8.** A May week comparison between a dwelling with one occupant (Nº2) (blue curve) and a dwelling of the same size with five occupants (Nº5) (red curve), concerning the conditions in the living room (LR): (**a**) temperature; (**b**) relative humidity.

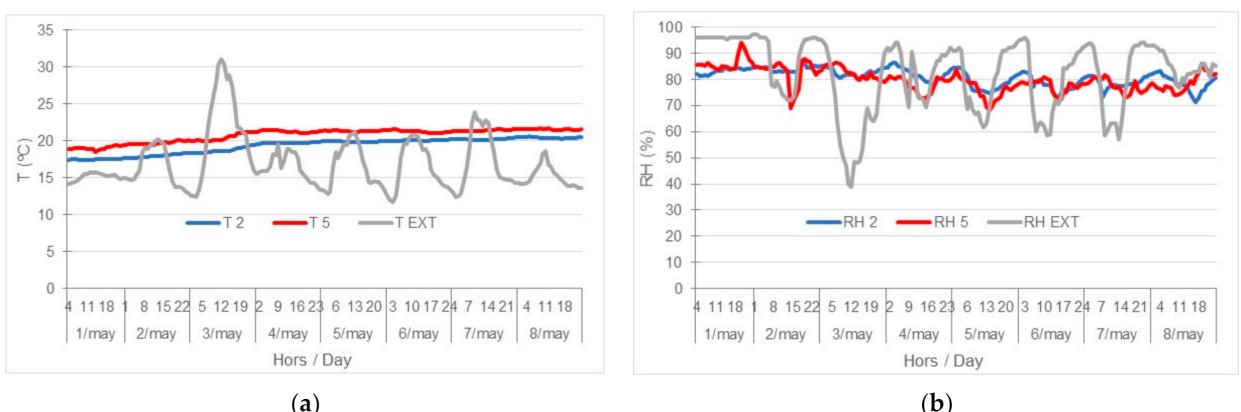

(**a**)          (**b**)

**Figure 9.** A May week comparison between a dwelling with one occupant (Nº2) (blue curve) and a dwelling of the same size with five occupants (Nº5) (red curve), concerning the conditions in the main room (R1): (**a**) temperature; (**b**) relative humidity.

The two presented cases demonstrate the discrepancy that can occur in the indoor conditions of a dwelling, depending on the type of occupation and use. As such, they justify the importance of simulations that consider the influence of the occupants.

*3.2. Social Survey*

The occupants of the monitored houses were asked to fill the social survey. The households of those six dwellings comprising a total of 15 occupants (almost 20% of the total neighborhood). Figure 10 shows the demographic characterization of the households that were asked to fill the survey. In spite of the relatively small coverage of the surveys, the distribution and trends identified by it match the empirical data gathered along the time by the social worker who gives support to the social neighborhood on a daily basis. Thus, the results of the survey were considered representative enough.

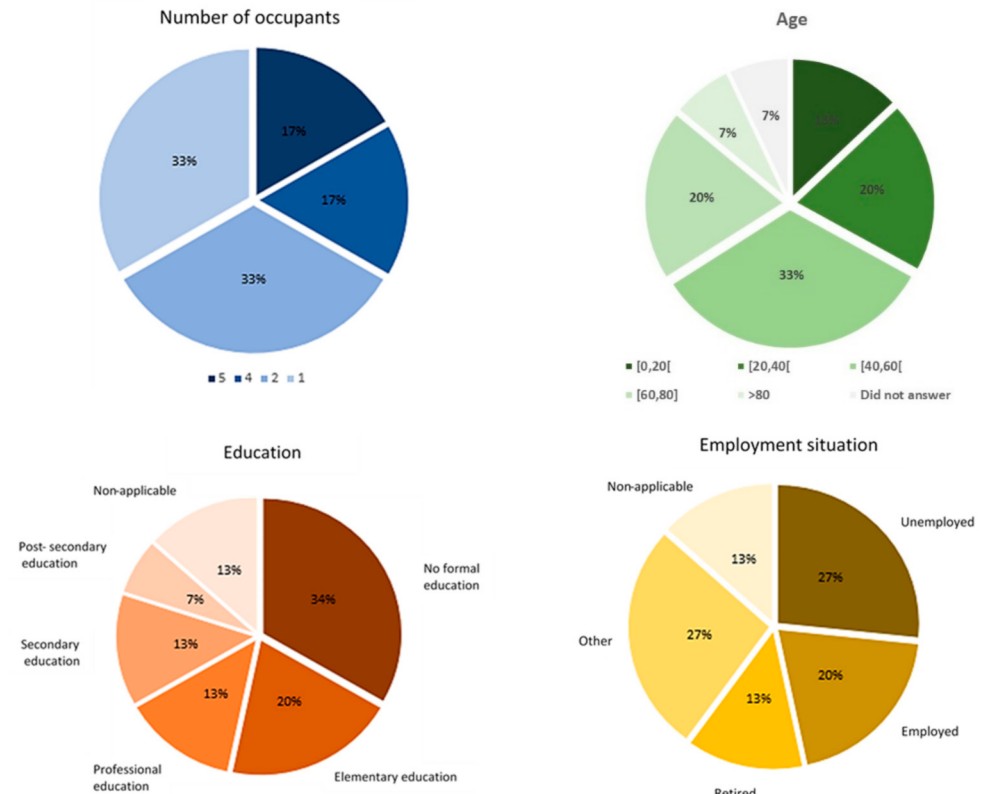

**Figure 10.** Demographic characterization of the studied neighborhood.

The occupants are distributed through all age groups, with the highest proportion of them between 40 and 60 years old. Regarding education, there is a large percentage of residents with no formal education or only with elementary education, and the situation of unemployment is the predominant reality.

The survey showed that most of the dwellings have fewer occupants than their maximum capacity, except for a dwelling that is overcrowded. Except for the dwellings with only one resident, all the dwellings have always at least one occupant at home the whole day. Although the survey was carried out in a period of confinement due to COVID-19, the authors believe that the usual presence of at least one occupant is mainly due to the unemployment rate and the retirement condition, since the respondents were asked to give their answers, taking into account the usual and not the exceptional situation.

In about 67% of the dwellings, there are construction anomalies, whether they are related to dampness or cracking.

All the respondents feel thermal discomfort; they consider their home too hot or too cold. All the respondents ventilate their homes by opening windows. Only 33% of residents heat their home, with portable heaters and just occasionally, and 17% use portable fans for cooling, also occasionally. All other occupants wear extra clothing or accept the feeling of discomfort in the heating season. In the cooling season, they choose to open windows or use shading devices, such as blinds.

Finally, 83% of the respondents experience difficulties in paying their electricity bills, 67% avoid the use of heating or cooling devices, and 83% avoid the use of other domestic devices and artificial lighting because of electricity consumption. These results point out a clear situation of energy poverty.

## 4. Discussion of Post-Renovation Monitoring and Use of Renewable Energies

### 4.1. Simulations

Using the Design Builder model, the buildings with the proposed new construction solutions for the external walls and roof were simulated throughout the whole year, using the reference year climatic data and considering a free-floating regime (no heating or cooling was provided).

To simplify the presentation of the results, two reference weeks are selected, where the highest and lowest outdoor temperature occur, for the cooling and heating season, respectively. Thus, the days chosen are from 25 January to 1 February and from 6 to 12 July, since they include the two peaks shown in Figure 11.

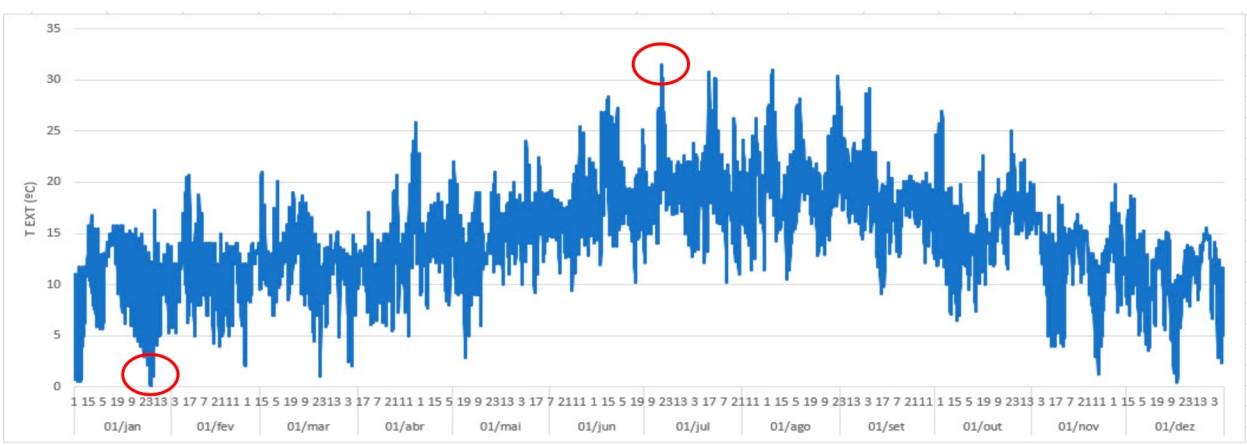

**Figure 11.** Outdoor temperature for the reference year.

### 4.2. Free Float Pre- and Post-Renovation Analysis

In order to make a comparison of the thermal behavior between pre- and post-renovation, dwellings Nº1, Nº2, and Nº6 were selected, since they were previously monitored and the real occupancy schedule could be used.

The graphs with the comparisons for the reference weeks are shown in Figures 12–14. In the reference week of the heating season, there is a significant improvement of the thermal performance of the building. For the dwellings Nº1, Nº2, and Nº6, there is a maximum temperature increase of 2.5, 3.0, and 3.5 °C, and average temperature increase of 1.0, 1.0, and 1.5 °C, respectively. However, in the cooling season, for dwellings Nº1 and Nº2, there was a slight rise in temperature, which, in this case, is unfavorable, since it increases the risk of overheating. In the cooling season, the increase in maximum temperature recorded before and after renovation is 1.5 °C for the dwelling Nº1 and 1.7 °C for the dwelling Nº2. Regarding the Nº6 dwelling, the planned renovation measures also result in an improvement of its thermal performance in the cooling season, with a maximum temperature reduction of 2 °C. This improvement in the dwelling Nº6 was expected, since this dwelling in its original conditions has quite high thermal gains through the roof in summer and, therefore, with the roof insulation, a great reduction of these gains is achieved.

In Figures 12–14, it can be seen that, for the free-floating simulation, post-renovation temperature values do not comply with the upper limit value (25 °C), nor with the lower limit value (18 °C), suggested as reference by the Portuguese regulation [32]. This means that simple renovation measures alone do not guarantee adequate comfort levels, although they improve indoor thermal conditions. It is important to note that the presented results correspond only to two extreme weeks of winter and summer. In many other periods during the year, comfort may be achieved through passive measures. Particular attention needs to be paid to summer comfort, since the application of thermal insulation increases the risk of overheating in some dwellings. However, there are ways of compensating that increase, such as improving night ventilation.

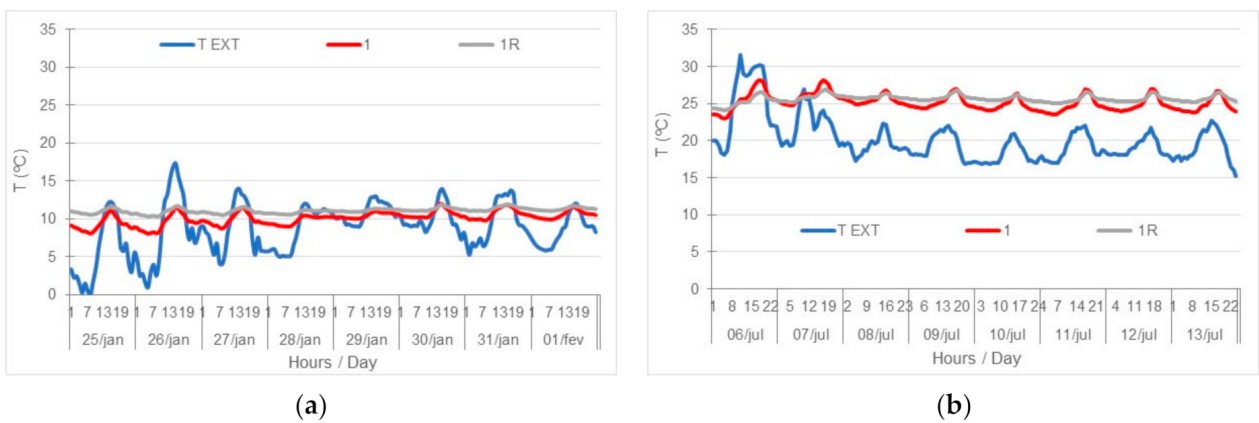

**Figure 12.** Comparison between temperatures before (red curve) and after (grey curve) renovation in dwelling Nº1: (**a**) January; (**b**) July.

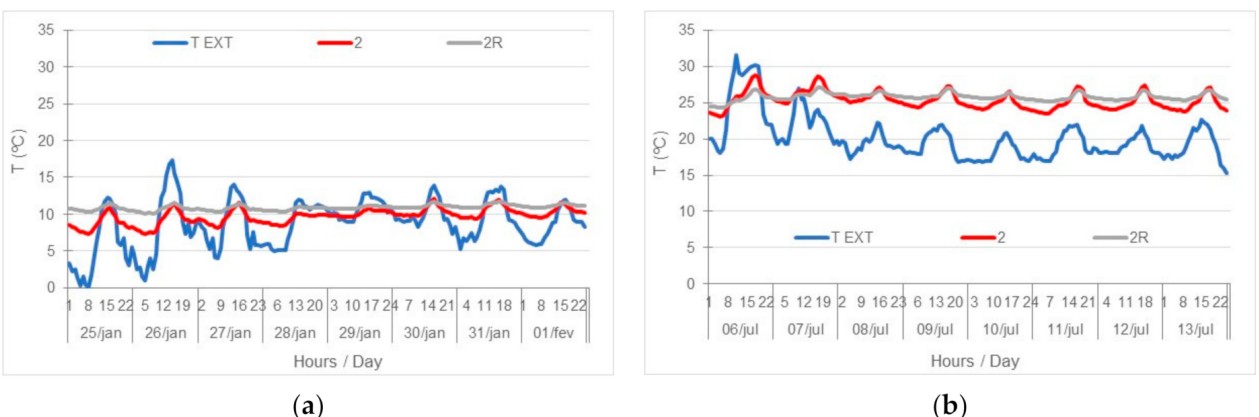

**Figure 13.** Comparison between temperatures before (red curve) and after (grey curve) renovation in Nº2: (**a**) January; (**b**) July.

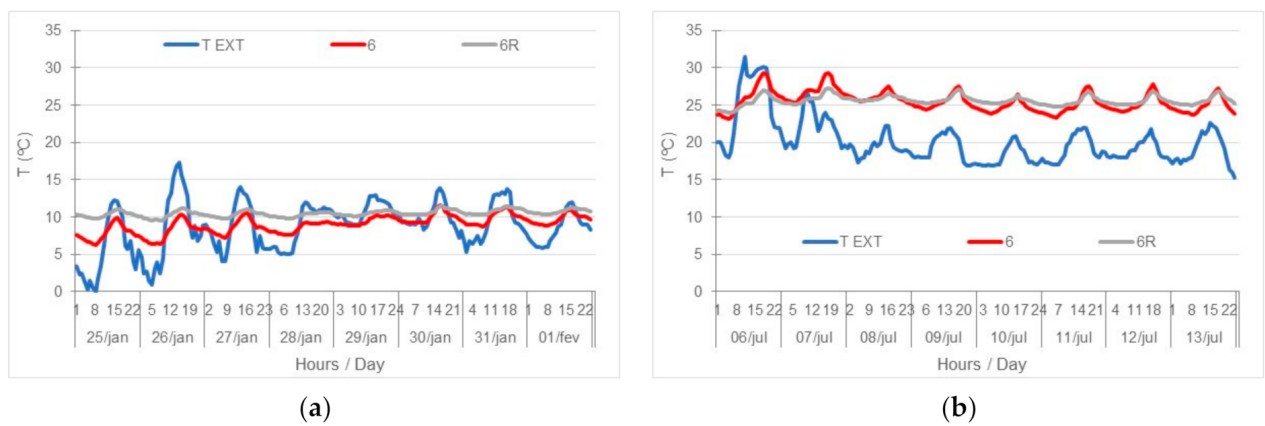

**Figure 14.** Comparison between temperatures before (red curve) and after (grey curve) renovation in Nº6: (**a**) January; (**b**) July.

### 4.3. Selection of the Dwellings to Be Monitored

#### 4.3.1. Preliminary Analyses

In order to select the dwellings to be monitored in the post-renovation plan, three analyses are carried out, where different dwellings are compared in relation to their orientation, location in the building, and between what are considered the most and the least critical dwellings.

#### 4.3.2. Influence of the Orientation

In this neighborhood, all the dwellings have windows facing east and facing west. Thus, the influence of the orientation was checked in the only possible way, as explained below. The dwellings chosen to be compared in terms of orientation were the following: Nº1 with the dwelling in the opposite side of the building at the same level—Nº8, Nº3 with Nº2, and Nº6 with its opposite Nº9. Although the glazed areas of these pairs of dwellings have the same orientations (both have windows on the east and on the west facades), there is a "blind facade" facing South (in the case of Nº3, Nº 8, and Nº9) and the opponents with a "blind facade" facing North (Nº1, Nº2 and Nº6).

By analyzing the resulting graphs (Figures 15–17), it can be concluded that the most vulnerable entrance is the one with the "blind facade" facing North in winter; this means its dwellings are colder than the ones of the southern entrance, in the heating season. Dwellings Nº1, Nº2, and Nº6 have on average temperatures lower 1.0, 1.2, and 1.3 °C, respectively. In the cooling season, there are no significant differences between the two sets of dwellings, as could be expected.

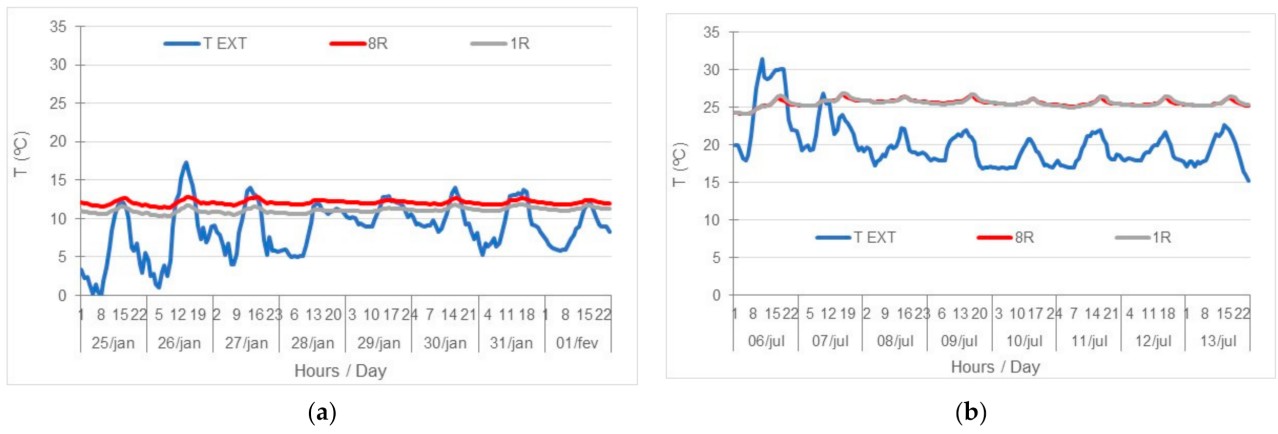

(a)　　　　　　　　　　　　　　　　　　　(b)

**Figure 15.** Comparison between temperatures in dwellings Nº1 (grey curve) and its opponent (red curve): (**a**) January; (**b**) July.

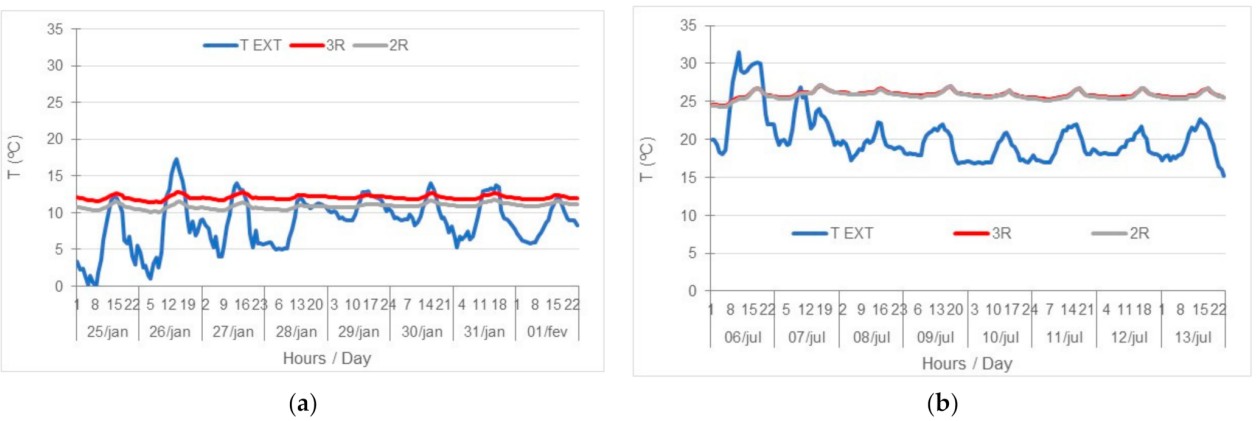

(a)　　　　　　　　　　　　　　　　　　　(b)

**Figure 16.** Comparison between temperatures in dwellings Nº3 (red curve) and Nº2 (grey curve): (**a**) January; (**b**) July.

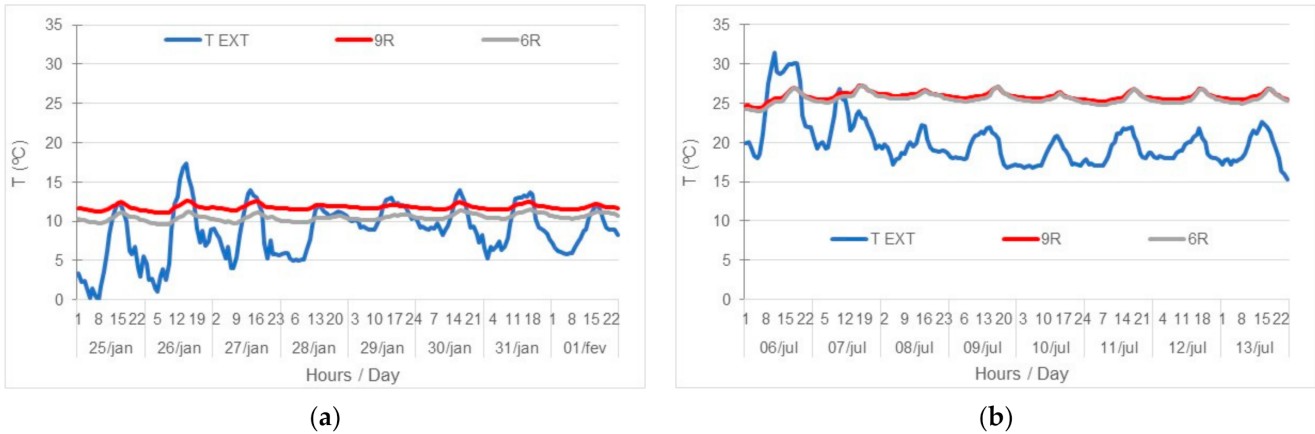

(**a**)  (**b**)

**Figure 17.** Comparison between temperatures in dwellings Nº6 (grey curve) and its opponent (red curve): (**a**) January; (**b**) July.

### 4.3.3. Vertical Location in the Building

To compare the vertical location of the dwellings in the building, the dwellings located in the most unfavorable entrance were chosen. The most unfavorable entrance was considered the one in which the dwellings have a "blind facade" facing North (Nº1, Nº2, and Nº6).

Examining the graphs, in the heating season, the Nº6 dwelling is the most critical, having the lowest temperatures, and this can be explained by the higher thermal losses through the roof, since this dwelling is located immediately beneath the roof. Regarding the cooling season, all dwellings behave similarly, in spite of the heat gains through the roof in Nº6, because, after renovation, those gains are very much reduced (Figure 18).

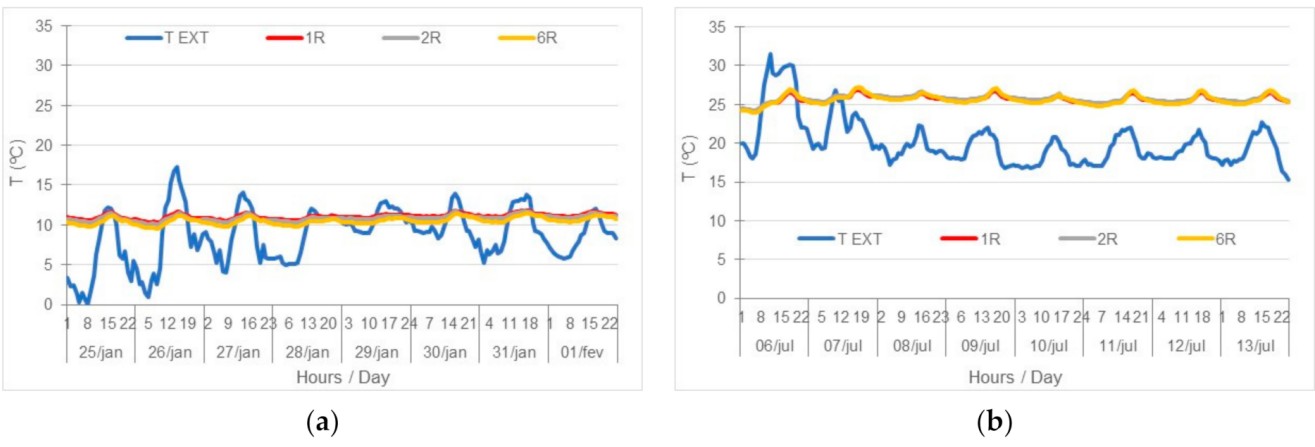

(**a**)  (**b**)

**Figure 18.** Comparison between temperatures in dwellings Nº1 (red curve), Nº2 (grey curve), and Nº6 (yellow curve): (**a**) January; (**b**) July.

### 4.3.4. The Most and the Least Critical Dwellings

Based on the two analyses made previously, it can be said that the Nº6 dwelling is the most critical in the heating season and will be compared with the least exposed dwelling of the same building, which means the one in a central location, vertically and horizontally, Nº10.

The graphs of Figure 19 show that the least exposed dwelling is warmer in both seasons, with a maximum temperature difference of 1.8 °C and an average difference of 1.6 °C in the heating season and 1.7 and 1.4 °C, respectively, in the cooling season, being a dwelling with a higher risk of overheating in the post-renovation. Since overheating is

related to thermal gains but also to losses, this dwelling has less chance of cooling, since it is less exposed to the outside than the dwelling N°6.

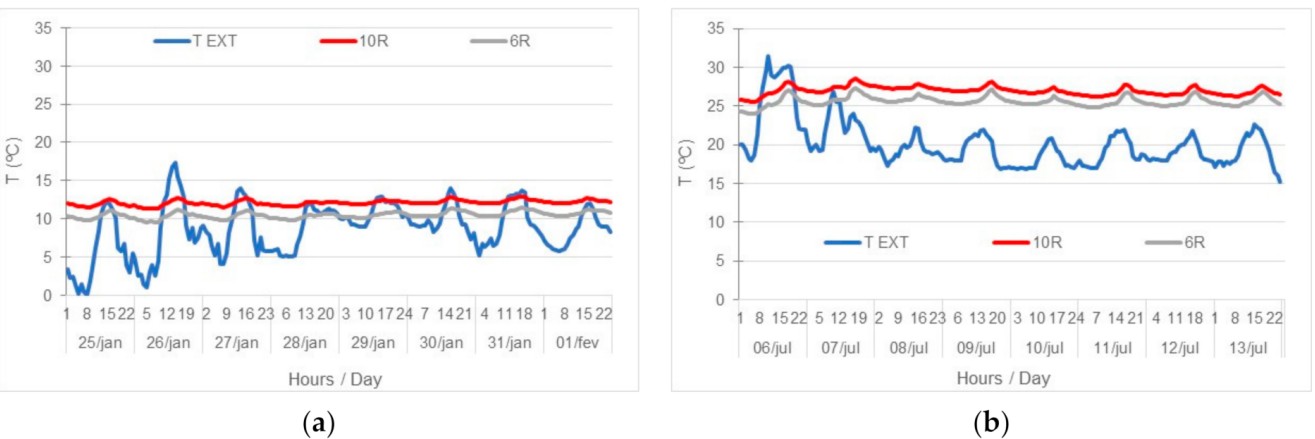

(**a**)                                                                                     (**b**)

**Figure 19.** Comparison between temperatures in dwelling N°6 (grey curve) and the least exposed dwelling (red curve): (**a**) January; (**b**) July.

In the post-renovation period, the dwelling N°6 will be the most critical dwelling in the heating season, and the least exposed one (N°10) will be the least critical one in the heating season but somehow problematic in the cooling season.

### 4.4. Pre and Post-Renovation Analysis Considering Heating Systems and Renewable Energies

The social–economical specificities of a social neighborhood, emphasized by the energy poverty present in these type of houses in Southern European countries, implies a customized approach in the renovation design. The energy poverty implies the use of passive design, but there is an extra need to supply all the heating and cooling needs by renewable energies. The surveys showed that all the occupants feel thermal discomfort, but the occupants only use occasionally heaters or electric fans to cool the environment. A simulation on the east block was made in order to estimate the energy demand using a heating system during the most frequent schedule with occupancy. The set-point for heating was 18 °C. The heating equipment considered was an electrical heater device with an efficiency of 1, because no interior renovation works are predicted in the renovation plan. As showed in Figures 4 and 5, considering the thermal comfort adaptive standard EN-15251 [33], the dwellings were far below the upper limits of the thermal comfort. Therefore, no cooling was considered in this simulation scenario. In order to supply the highest amount possible of the electrical demand, the roof was covered with photovoltaic panels (PV). Each PV had a power peak of 270 W and an active area of 1.42 m$^2$. In total, 112 individual PV panels were simulated. Table 4 shows that the renovation works on the passive side of the building would reduce the energy demand for heating from 57,029.4 to 39,070.5 kWh (−31.5%). The energy balance was obtained through the dynamic simulation software. In spite of the proximity of the energy generated by roof PVs compared to the electric demand in the post-renovation scenario (49,346.2 and 70,439.0 kWh, respectively) the hourly balance of energy leads to an electrical energy purchase from the grid of 60,218.0 kWh and to an injection of 39,125.2 kWh in the grid (surplus) (Table 5). The above values are affected by the option of having an occupancy-based heating schedule that is turned off some hours during the daylight (PV peak production). Considering a scenario with 24 h of heating (conditioned by the set-point of 18 °C), the overall heating demand would increase 8340 kWh, but only increasing about 3240 kWh in the electricity purchased from the grid; the rest comes directly from the PVs (5100 kWh). Therefore, there is a need to use other renewable sources or storage systems in order to enable thermal comfort to the occupants of this social neighborhood.



**Table 4.** East building energy loads considering pre- and post-renovation scenarios.

| Data | Lighting and Room Electricity (kWh) | DHW (Gas) (kWh) | PV Generation (kWh) | Pre-Renovation Simulation | Post-Renovation Simulation |
|------|------|------|------|------|------|
| | | | | Heating (kWh) | Heating (kWh) |
| Jan | 2656.7 | 1900.5 | 2364.0 | 13,772.0 | 9625.2 |
| Feb | 2406.5 | 1733.9 | 2957.0 | 9631.0 | 6696.5 |
| Mar | 2674.6 | 1945.3 | 4499.9 | 7091.0 | 4926.0 |
| Apr | 2573.3 | 1844.9 | 5198.6 | 2991.4 | 2164.7 |
| May | 2656.7 | 1900.5 | 5452.0 | 1190.1 | 775.3 |
| Jun | 2591.2 | 1889.7 | 5499.4 | 21.1 | 17.2 |
| Jul | 2656.7 | 1900.5 | 5479.1 | 0.0 | 0.0 |
| Aug | 2665.7 | 1922.9 | 5474.0 | 0.2 | 0.0 |
| Sept | 2582.3 | 1867.3 | 4481.3 | 51.9 | 3.9 |
| Oct | 2656.7 | 1900.5 | 3717.9 | 2340.0 | 1294.8 |
| Nov | 2582.3 | 1867.3 | 2094.0 | 8155.3 | 5371.7 |
| Dec | 2665.7 | 1922.9 | 2129.0 | 11,785.4 | 8195.2 |
| TOTAL | 31,368.5 | 22,596.1 | 49,346.2 | 57,029.4 | 39,070.5 |

**Table 5.** East building surplus and purchased electrical energy considering the post-renovation scenario.

| Data | Surplus kWh | Purchased kWh |
|------|------|------|
| Jan | 1686.6 | 11,604.5 |
| Feb | 2138.7 | 8284.8 |
| Mar | 3405.2 | 6505.9 |
| Apr | 4189.6 | 3728.9 |
| May | 4393.0 | 2373.1 |
| Jun | 4452.8 | 1561.8 |
| Jul | 4514.1 | 1691.7 |
| Aug | 4562.8 | 1754.5 |
| Sept | 3755.8 | 1860.7 |
| Oct | 3030.7 | 3264.3 |
| Nov | 1484.6 | 7344.6 |
| Dec | 1511.3 | 10,243.1 |
| TOTAL | 39,125.2 | 60,218.0 |

Figure 20 shows the energy consumptions with heating of the dwellings N°1, N°2, N°6, and N°10, considering the pre- and post-renovation scenarios. In line with the observation in the previous section, it is possible to see the vertical location in the building influence on the energy consumption of the dwellings. Since one of the renovation actions was the thermal insulation of the attic, the dwelling N°6 was the one with the most evident decrease in heating energy consumption after the renovation. Dwelling N°10, as it is the dwelling located in the most favorable location of the building in the heating season, had the lowest change, in percentage, with the renovation.

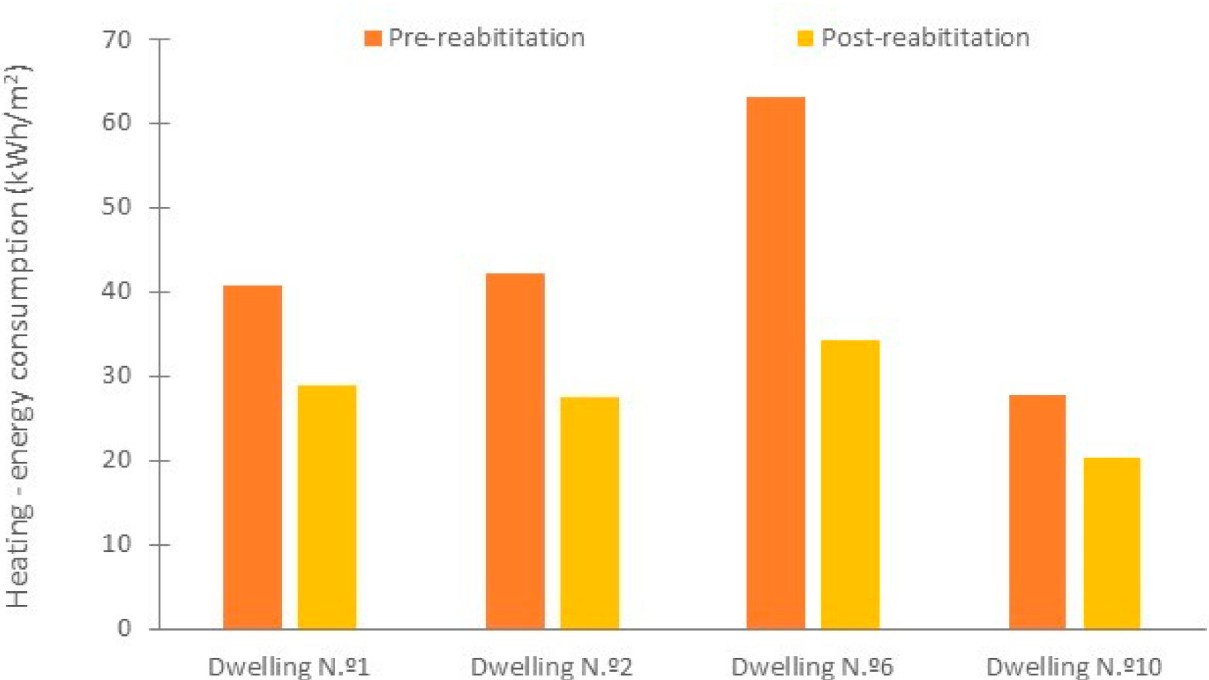

**Figure 20.** Energy consumption of the heating system in the dwellings number 1, 2, 6, and 10 in kWh/floor area.

## 5. Recommendation for Post-Renovation Monitoring and Renewable Energies

*5.1. Recommendations on Monitoring*

The recommendations on monitoring are divided into two groups: the architecture of the monitoring system, and the selection of dwellings and sensor placement at the dwellings' scale and rooms' scale and its operation.

### 5.1.1. Monitoring System Architecture

In the present study, some data were lost. Since the present study was carried out in a period of confinement due to COVID-19, the visits to the monitored dwellings were not possible. The data extraction of the used monitoring system was wireless, functioning via BLE (Bluetooth low energy). However, to reduce the energy consumption of the sensors, the BLE had to be activated, pushing a button present in the sensor. Although the estimated autonomy of the battery of the used sensor is about one year, no information about the battery status was available. The constraint imposed by the pandemic of COVID-19 highlighted the need to have platforms to fully access the data remotely.

The proposed monitoring system architecture is based on individual devices with sensors of temperature and relative humidity in every room. The knowledge discovered from each sensor differs from the room type. It could be used to determine thermal comfort in the main room and the risk of condensation and health-related risks in all the compartments. The system architecture proposal is represented in Figure 21. The individual sensors should be wirelessly linked with an intermediate access point that should be connected to a router that sends the data to a cloud data warehouse. To reduce the internet services costs, it is possible to use only one router per building and intermediate access points wired to it. A platform should be created to access the data stored and constantly analyze the data in real-time. The advantage of this methodology is to have multidisciplinary alerts, given the thresholds as reference values, notifying the operator if outlier values are being registered. The advantages of this solution would be the possibility of accessing information whenever necessary, controlling errors almost immediately, and less disruption to occupants due to less frequent visits. The future renovation design will also be more data-driven, and the maintenance operations could be more predictive. A

disadvantage would be the energy consumption of the sensors that should be connected to the electrical network. However, the energy consumption of temperature and relative humidity sensors mounted on an Arduino platform is very low (<<1 Wh). On the other hand, this architecture allows continuous monitoring without visiting the interior of the dwellings and makes possible the inclusion of another type of sensors, with higher energy consumptions. The inclusion of these sensors is not possible with an energy supply guaranteed by batteries, and these sensors could be used to increase the knowledge on IEQ like $CO_2$, $PM_{2.5}$, and TVOC sensors [24,35]. It is also important that the energy and water consumptions of each dwelling are measured by energy/water meters, and the information should be supplied to the intermediate access points of each floor.

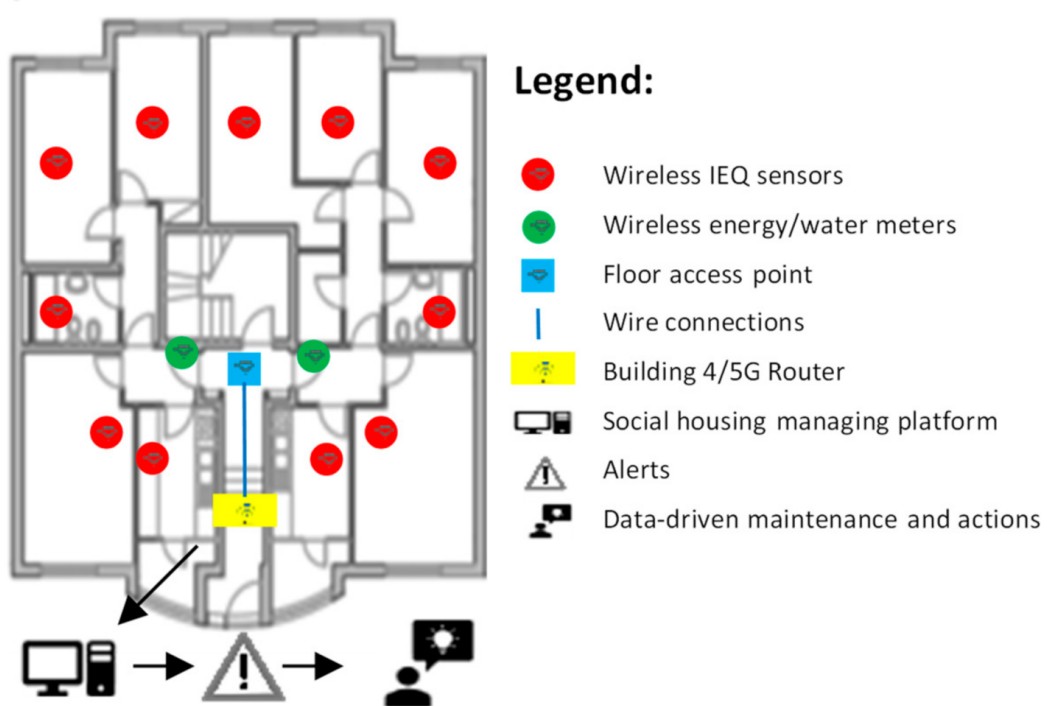

**Figure 21.** Proposed monitoring system architecture.

5.1.2. Selected Dwellings, Sensor Placement, and Operation

A monitoring system is considered more effective if redundant information is provided. Furthermore, occupants' operation of the building systems is known to have a great influence on the energy and IEQ conditions. Therefore, and considering that the dwellings selected for the pre-renovation monitoring campaign consisted in the dwellings facing the extreme exterior conditions, the proposal for post-renovation monitoring involves keeping the monitoring of all the dwellings previously monitored with some updates. This option would enable the comparison between the pre- and post-renovation hygrothermal conditions.

The proposed dwellings to be monitored include the upper dwelling with the largest facade facing the North; dwellings in alignment with the previous but in intermediate and ground floors; a dwelling in the center of the building (with extreme conditions in the cooling season); the dwelling with the highest density of occupation; a dwelling with no occupancy or, if none exists, a dwelling with the lowest occupancy density; and a dwelling in the opposite side of the alignment previously proposed.

If there are some limitations in the number of sensors, at least three dwellings must be monitored: the dwelling in the upper floor and with the largest façade area facing North, a dwelling in the center of the building in height and width, and a dwelling in the same conditions as one of the before mentioned but with different occupancy density.

The placement of a sensor in each main room is essential, in order to have a complete analysis of the hygrothermal conditions of the dwellings. Its placement should be based on the requirements followed in the pre-renovation campaign, positioned in the breathing zone, and glued to the partition wall of the room oriented to the North, when possible, avoiding proximity from electrical equipment.

In order to avoid monitoring errors, it is advisable to collect data frequently, so that there is no loss of data and a continuous data analysis is performed. In the present case, the logging interval of 15 min was considered ideal, not because the data were mined in that interval, but because it was useful to reduce the faulty values to obtain the hourly means. Therefore, logging intervals of 15 min or below are recommended. Another recommendation in order to avoid measurement errors and to make the installation visits of the monitoring plans as efficient as possible is to carry out the configuration and testing of the sensors in the laboratory in advance.

The monitoring on site of the outdoor climate, temperature, and relative humidity is also important to obtain the water vapour excess. The collection of solar irradiance values from a local meteorological station would contribute to get more reliable information on heat gains.

Carrying out another survey after renovation will also be relevant to know if any important change in occupancy or type of use has occurred between the pre- and post-renovation scenarios. As suggested by participatory actions Lucchi and Delera [25], surveys in the pre-renovation phase should include the perspective of occupants to enable a user-centric design proposal in an early stage. The post-renovation survey should also assess the degree of satisfaction of the occupants with the result of the renovation works, regardless of the values measured for the physical parameters.

*5.2. Recommendations for Renewable Energy in Social Housing*

As it was shown, the thermal comfort reference conditions of the Portuguese regulation are unlikely to be reached due to the social–economic specificities of the inhabitants of the social neighborhoods in Portugal. The energy poverty is a reality that can be explained by the low income of these families, often aggravated by a high unemployment rate (27%, in the case of the studied neighborhood, Figure 10). The simulated results highlighted that in the heating season, the average indoor temperature was around 12 °C in the dwelling with the most favorable boundary conditions and near 10 °C in the dwelling with the less favorable ones (Figure 19). The passive measures are not enough to ensure thermal comfort. Therefore, in order to meet thermal comfort minimum requirements, renewable energies could be included in the renovation plan. The total use of the roof of the East block of the present case study for PVs would generate 49,346.2 kWh. However, in the post-renovation scenario, only 10,221 kWh would be directly used by the building, and the rest would have to be injected in the electrical grid or storage. Therefore, in the present case, using PVs installed on a surface of around 1/3 of the dwellings floor area, the surplus could be injected in the grid with estimable profits of around 1565 € per year (≈0.04 €/kWh). However, to fulfil the heating, lighting, and appliances demand, the building has to purchase 60,218 kWh from the grid at an average price of 0.212 €/kWh (Portuguese household consumers final price in 2020 [36]). The 39,125 kWh injected in the grid will enable the purchase of 7382 kWh from the grid. This scenario can be even worse if no energy trader is interested in buying the PVs surplus, in which case energy is injected into the grid free of charge. The use of energy storage systems could be considered to increase the use of the energy generated by the PVs. Furthermore, given the energy poverty of the neighborhood inhabitants, if comfort is to be met, the remaining energy needs should be covered by renewable energy systems able to supply energy beyond the solar daylight schedule.

## 6. Conclusions

The specificities of the inhabitants of social housing in Portugal require that renovation plans be customized. This work proposes requirements to be fulfilled by a monitoring plan to be implemented in the frame of a renovation program, and it includes both pre- and post-intervention phases.

The results of monitoring hygrothermal parameters detected the existence of temperature values frequently below 18 °C, the heating season reference value, but no significant overheating was observed in the cooling season. Comparing identical dwellings with different occupancy profiles, the over-occupied dwelling presented an average temperature 1.2 °C above the under-occupied dwelling. In the cooling season, the comparison of two identical dwellings, one occupied and other unoccupied, revealed an average temperature greater in approximately 1 °C of the unoccupied dwelling. These findings highlighted the specificities of the occupants and the different impacts that they could have in the indoor environment. The passive renovation plan allows the occupants' energy needs to be reduced considerably by approximately halving heating needs. Despite the proposed renovation action, thermal comfort throughout the year will not be ensured by passive means alone. The proposed monitoring and survey plan allows occupants to be involved in a design phase so that some of their needs can be met. On the other hand, the monitoring plan makes it possible to reduce the sampling to only a few control dwellings, considered with extreme boundary conditions in the heating and cooling seasons. The monitoring plan will allow the construction of more reliable monitoring systems in order to increase the necessary knowledge for the elaboration of occupant-centric architecture and engineering designs, as well as to confirm the effectiveness of the predictive simulations.

The case study presented in this work highlights several practical restrictions that must be attended in a social housing renovation process. To prevent a reallocation of the occupants during renovation works, the intervention is to be limited to exterior areas of the dwellings. As a complement to the renovation of the opaque envelope, a study of the potential use of PVs revealed a potential reduction of the need to purchase energy by 15%. However, this value could increase with the use of energy storage systems, since 79% of the total PVs electrical energy generated was surplus. Due to the typical energy poverty scenario that was observed in the case study, the improvement of thermal comfort conditions should be focused on an increase of renewable energy use. In this specific case, the inclusion of battery energy storage systems (BESS) connected to the PVs, in addition to alternative renewable energies with the capacity of producing energy beyond the solar daylight schedule, is a possibility to be explored in further developments of the ongoing SUDOE ENERGY PUSH (SOE3/P3/E0865).

In the social housing context, if no renewable energies are included in renovation plans, the focus of low energy renovation should be first to improve occupants' comfort instead of reducing energy consumption, since, in practice, heating and cooling is punctual or non-existent. Improving comfort by passive means leads to a reduction in energy needs. If the remaining reduced needs are supplied by renewable energy sources, then energy poverty will be largely overcome, and an enhanced quality of life will be provided to the most vulnerable population.

**Author Contributions:** Conceptualization: P.F.P., H.C. and N.M.M.R.; methodology: H.C. and P.F.P.; software: B.S.; validation: H.C. and P.F.P.; formal analysis: H.C. and N.M.M.R.; investigation: B.S., P.F.P. and H.C.; resources: P.F.P., H.C. and C.P.; data curation: B.S.; writing—original draft preparation: B.S.; writing—revision and editing: P.F.P., H.C. and C.P.; visualization: B.S. and P.F.P.; supervision: P.F.P. and H.C.; project administration: N.M.M.R.; funding acquisition: N.M.M.R. All authors have read and agreed to the published version of the manuscript.

**Funding:** This research was funded by Base Funding—UIDB/04708/2020 of the CONSTRUCT—Instituto de I&D em Estruturas e Construções funded by national funds through the FCT/MCTES (PIDDAC) and by SUDOE ENERGY PUSH (SOE3/P3/E0865), which is a project co-funded by the Interreg Sudoe Programme through the European Regional Development Fund (ERDF).

**Institutional Review Board Statement:** Not applicable.

**Informed Consent Statement:** Not applicable.

**Data Availability Statement:** The data are not publicly available due to privacy restrictions.

**Acknowledgments:** The authors would like to thank Gaiurb EM, the municipal company from Vila Nova de Gaia company responsible for Urbanism, Social Housing, and Urban Renovation for their participation in this research project, input, and availability to discuss the results.

**Conflicts of Interest:** The authors declare no conflict of interest.

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
