# Peer review of "Low Energy Renovation of Social Housing: Recommendations on Monitoring and Renewable Energies Use"

_sustainability, doi:10.3390/su13052718_

Round 1
Reviewer 1 Report
The paper concerns the renovation of social housing, suggesting a monitoring study and a survey for the evaluation of the actual conditions. The case study is located in the North of Portugal. Indoor hygrothermal conditions are analysed to identify the differences in indoor conditions of the dwellings and understand the influence of occupancy density and occupants’ behaviour. The first part of the introduction untile row 99 is really interesting and well focused in the topic. I suggest to focus better on the gap in literature and on the novelty of your study. Several studies have been done on social aspects and monitoring of social housing neighborhoods, considering energy consumption, poverty and human inclusion and cohesion as focus of the study. In the fist case, the study refer to the knowledge of human behavior as well as on the engagement of people in a user-driven design approach. In the second case, they focused on energy and environmental monitoring. I would suggest a similar experience in a social housing in Milan in Italy to engage people in the design with students the renovation and the energy retrofit of the historical neighborhood, considering a socio-ecological approach that connect people, energy, use of resources, transportation. Some more information are provided by the paper https://doi.org/10.3390/buildings10090159. I think that this can enlarge you horizon, demonstrating that your study is important also for social and experimental aspects. Here you can find also several new paper in this topic. I suggest to move the part From row 110 in the methodology. In the methodology is not clear you approach. First, you did a social survey and a monitoring Then you suggest how to define a monitoring plan. Is it true? Why not a survey plan? Regarding the social survey, in the methodology, explicate better the approach: how many people you involve? In which way you select them? How many people live there? It is a significant rate? Please, support your approach by a statistical scientific approach. Now, it is not scientific. Explicate better your results giving a key for interpretate it. The novelty of you approach is not clear. Explicate better why monitoring is needed and which more information it can give to you redevelopment plan. Conclusion must be completely revised.
Author Response
The paper concerns the renovation of social housing, suggesting a monitoring study and a survey for the evaluation of the actual conditions. The case study is located in the North of Portugal. Indoor hygrothermal conditions are analysed to identify the differences in indoor conditions of the dwellings and understand the influence of occupancy density and occupants' behaviour.
The first part of the introduction untile row 99 is really interesting and well focused in the topic. I suggest to focus better on the gap in literature and on the novelty of your study. Several studies have been done on social aspects and monitoring of social housing neighborhoods, considering energy consumption, poverty and human inclusion and cohesion as focus of the study. In the fist case, the study refer to the knowledge of human behavior as well as on the engagement of people in a user-driven design approach. In the second case, they focused on energy and environmental monitoring. I would suggest a similar experience in a social housing in Milan in Italy to engage people in the design with students the renovation and the energy retrofit of the historical neighborhood, considering a socio-ecological approach that connect people, energy, use of resources, transportation. Some more information are provided by the paper https://doi.org/10.3390/buildings10090159. I think that this can enlarge you horizon, demonstrating that your study is important also for social and experimental aspects. Here you can find also several new paper in this topic.
R:Thank you for the positive feedback.
We agree with the reviewer's comment; the "Introduction" section was reformulated accordingly. We also thank the paper suggested, it is a very interesting work, and it was included in the state-of-the-art.
A new section was added to give more insights about the energy benefits of the renovations works.
I suggest to move the part From row 110 in the methodology. In the methodology is not clear you approach. First, you did a social survey and a monitoring Then you suggest how to define a monitoring plan. Is it true? Why not a survey plan? Regarding the social survey, in the methodology, explicate better the approach: how many people you involve? In which way you select them? How many people live there? It is a significant rate? Please, support your approach by a statistical scientific approach. Now, it is not scientific.
R:The survey was performed to know the occupancy profiles of the dwellings. The main objective of this study was to propose a monitoring plan for analysing the pre-retrofit conditions and the post-retrofit benefits in terms of thermal comfort. The main objectives of this study were rewritten in section "introduction". The methodology section was modified to clarify the scope of the surveys.
The Section "3.2 Social survey" was improved to introduce more information about the surveys.
Explicate better your results giving a key for interpretate it.
R:Section "3. Results" and section "4.4. Recommendations" were reformulated accordingly. We also thank the paper suggested, it is a very interesting work, and it was included in the state-of-the-art.
The novelty of you approach is not clear. Explicate better why monitoring is needed and which more information it can give to you redevelopment plan.
R: Last paragraph of Section 1 was modified accordingly. A new section (5) was created to highlight the recommendation for monitoring.
Conclusion must be completely revised.
R: We agree with the reviewer's comment: the "Conclusion" section was reformulated accordingly.
Reviewer 2 Report
This paper studies the energy renovation of social housing buildings which is very interesting and topical.
The instrumentation and real-time monitoring of temperature and humidity in these buildings with the solutions proposed provide real answers to the questions that are being asked today about the interest of renovating these buildings.
I have a remark regarding the figures, maybe use a good resolution to make them clearer, and also put the legend of the graphics for figures 4 and 5.
I think that this study should be continued and deepened by working in particular on other proposals for renovation materials, such as bio-based materials.
Author Response
This paper studies the energy renovation of social housing buildings which is very interesting and topical. The instrumentation and real-time monitoring of temperature and humidity in these buildings with the solutions proposed provide real answers to the questions that are being asked today about the interest of renovating these buildings.
R: Thank you for the positive feedback.
I have a remark regarding the figures, maybe use a good resolution to make them clearer, and also put the legend of the graphics for figures 4 and 5.
R: We agree with the reviewer's comment; the figures were reformulated accordingly.
I think that this study should be continued and deepened by working in particular on other proposals for renovation materials, such as bio-based materials.
R: Thank you for the suggestion. We will take that in account for future developments.
Reviewer 3 Report
Modifying the contents of Title :
Social Housing Buildings to Social Houses Buildings
keywords : social houses buildings, renovation, monitering, simulation(design builder)
- The purpose and significance of conducting this study in introduction should be presented more clearly.
Figure 1 shows two similar pictures. Replace Chapter 1 with a photo that shows the elevation of the building's outer surface.
Paragraphs 1.2, 1.3, and 1.4 are modified to 2.2, 2.3, and 2.4.
Need to explain what the plot represents in Fig. 4, 5 and also to display a legend for each graph line
in Fig 20; Need description for Proposed monitoring system architecture
in p9. ; Modify fig.2 to fig.7
It is not clear what this paper presents in its final form
In other words, more explanation of energy elements is needed before and after remodeling.
Author Response
Modifying the contents of Title and keywords
R: The title and keywords were changed taking into account your opinion
The purpose and significance of conducting this study in introduction should be presented more clearly.
R: We agree with the reviewer's comment; the "Introduction" section was reformulated accordingly.
Figure 1 shows two similar pictures. Replace Chapter 1 with a photo that shows the elevation of the building's outer surface.
R: Unfortunately, we do not have the suggested type of photo. However, Figure 1 was reformulated to better show the case study.
Paragraphs 1.2, 1.3, and 1.4 are modified to 2.2, 2.3, and 2.4.
R: Thanks, the typos were removed.
Need to explain what the plot represents in Fig. 4, 5 and also to display a legend for each graph line
R: We agree with the reviewer's comment; the figures were reformulated accordingly.
in Fig 20; Need description for Proposed monitoring system architecture
R: Figure 20 was changed accordingly.
in p9. ; Modify fig.2 to fig.7
R: Thanks, the typos were removed.
It is not clear what this paper presents in its final form In other words, more explanation of energy elements is needed before and after remodeling.
R: The construction solutions are detailed in Table 1 (current situation) and Table 2 (post-renovation).
A new section was added to give more insights about energy consumptions before and after the renovation.
Reviewer 4 Report
Following is the comment from the reviewer:
1) Why was a small social housing neighbourhood in the municipality of Vila Nova de Gaia examined? How does this represent the current housing stock?
2) How many people responded to the survey? What is the response rate?
3) Figure 10 provides the demographic characterisation of the survey. Did the authors invite young people from 0-18 years old to participate in the survey? 13% of [0-20] was provided in the Figure.
4) Why was the information related to Education and Employment situation collected? How do these affect the results?
Author Response
Following is the comment from the reviewer:
1) Why was a small social housing neighbourhood in the municipality of Vila Nova de Gaia examined? How does this represent the current housing stock?
R: The present work was carried out within the scope of the INTERREG SUDOE project. In the building stock managed by the Portuguese municipality involved in the project this neighbourhood was the only one that will be retrofitted this year. This particularity will be important in future works. The social housing stock is heterogeneous, but the conclusion of this work tried to be general and replicable to the housing stock of the three Southern European countries involved in the project.
Clarifications were included in section "2. Methodology".
2) How many people responded to the survey? What is the response rate?
R: The Section "3.2 Social survey" was improved to introduce more information about the surveys.
3) Figure 10 provides the demographic characterisation of the survey. Did the authors invite young people from 0-18 years old to participate in the survey? 13% of [0-20] was provided in the Figure.
R: The text that introduces Figure 10 was modified to clarify it.
4) Why was the information related to Education and Employment situation collected? How do these affect the results?
R: That information regarding the education was not used in the context of this paper. The information about the employment situation was used to understand the occupancy of the dwellings in the building energy simulation phase.
Round 2
Reviewer 1 Report
Thank you very much for considering my suggestions
Author Response
We would like to thank for the comments of the reviewer and the time taken in the analysis of the article.
Reviewer 3 Report
in p.10.; Modify fig.2 to fig.7
in p.21 : Modify 5. conclusions to 6. conclusions
Author Response
Thanks, the typos have been corrected.
Reviewer 4 Report
The comments from the reviewer were well-addressed. There is no further comment.
Author Response

(The authors gave the same response as above.)
